# Dynamic Low-rank Estimation for Transformer-based Language Models

**Ting Hua**[1*], **Xiao Li**[2*], **Shangqian Gao**[1], **Yenchang Hsu**[1], **Yilin Shen**[1], **Hongxia Jin**[1]

[1]Samsung Research America, [2]University of Michigan, Ann Arbor

[1]{ting.hua,s.gao1,yenchang.hsu,yilin.shen,hongxia.jin}@samsung.com
[2]{xlxiao}@umich.edu

## Abstract

Matrix decomposition methods, such as Singular Value Decomposition (SVD) and its importance-weighted variants, have been widely used for compressing Transformer-based language models. While importance-weighted decomposition methods alleviate the strong assumption of equal importance for each parameter in SVD, they still rely on two fundamental assumptions: 1) unchanged importance distribution during further fine-tuning, 2) equal importance across weight matrices in different layers. Furthermore, these methods necessitate a well-trained task-specific model as the starting point and require additional fine-tuning after compression. In this work, we proposed RankDyna, a matrix decomposition method that enables dynamic rank resource allocation among matrices across different layers during the training process. Starting from a general pre-trained model, RankDyna accomplishes the dual goals of compression and adaptation to the downstream task, all within a single round of fine-tuning. The extensive evaluations demonstrate that RankDyna can outperform current SOTA methods under various parameter budget levels, and the advantage of RankDyna is further enhanced with higher compression rates.

## 1 Introduction

Transformer-based language models (Devlin et al., 2018; Waswani et al., 2017) have obtained significant success in a variety of Natural Language Processing tasks. However, these models are usually pre-trained by the huge corpus with millions or even billions of parameters, and hard to be deployed to resource-constrained devices. Therefore, the compression of the Transformer-based language model have attracted extensive attentions.

The Transformer blocks are primarily built with linear layers, which will convert the input features into output features through a weight matrix. By

applying low-rank estimation methods (Golub and Reinsch, 1971; Noach and Goldberg, 2020), the large weight matrix in a layer can be decomposed into two smaller matrices, resulting in the creation of two reduced-size linear layers. Also, it is important to note that matrix factorization technologies are orthogonal to other compression approaches such as Knowledge distillation (Sun et al., 2019; Sanh et al., 2019; Jiao et al., 2019) or Quantization (Shen et al., 2020; Zhao et al., 2021).

The standard SVD aims to minimize the reconstruction error, which results in the removal of the portion associated with small singular values. However, previous work have revealed that this objective does not always align with the objective of preserving task performance (Hsu et al., 2022; Hua et al., 2022). This is primarily due to the fact that parameters have various influences on the overall performance, whereas the standard SVD treats all parameters as equally significant.

To tackle this issue, previous work FWSVD (Hsu et al., 2022) and TFWSVD (Hua et al., 2022) attempt to conduct low-rank estimation on the matrix weighted by Fisher information. These approaches need an already fine-tuned model as their starting point to calculate importance scores based on Fisher information, and often require an additional round of fine-tuning to attain better performance. Besides, they are based on the assumption that the Fisher information of parameters, computed from a fine-tuned task-specific model, can accurately represent the importance distributions of those parameters. Additionally, they also assumed that these importance distributions remain unchanged throughout the additional round of fine-tuning.

In this paper, we propose a novel method called RankDyna, which conducts low-rank estimation throughout the fine-tuning process. Starting with a full-rank matrix, RankDyna gradually reduces the parameter budget level using a scheduler. At each step, the least important singular groups are

---

*These authors contributed equally to this work.

removed based on their importance scores, which are determined by the amount of impact they contribute to the loss. RankDyna collectively evaluate the importances of all singular groups within the model, and therefore capable of assigning different ranks to each matrix. The advantages of the proposed RankDyna can be summarized as follows:

**Dynamic tracking of importance for singular groups**. Previous importance-aware decomposition approaches aimed to optimize an objective weighted by static importance assigned to each parameter. However, assuming that the importance distribution of parameters remains unchanged during fine-tuning is unrealistic. Unlike these methods, RankDyna can capture the dynamic changes in importance associated with singular groups, thus improving performance compared to the methods based on the static importance assumption.

**Better parameter allocation accross matrices**. The varying sensitivities of different layers in Transformer models during compression are well-established. Given a targeted model size budget, previous work on low-rank estimation will assign uniform ranks to all layers. In contrast, RankDyna enables dynamical parameter allocation across layers by monitoring their importance variations, leading to significantly better compression outcomes.

**Reduced computational and storage resources**. Unlike previous methods that rely on already fine-tuned task-specific models, our approach begins with a general pre-trained model. As a result, RankDyna only needs a single round of fine-tuning. This significantly reduces the computational overhead compared to methods that require two rounds of fine-tuning. Previous approaches require factorization for each fine-tuned task-specific model that can be increasingly expensive when model size gets larger[1]. In contrast, RankDyna only requires a one-time factorization on the base model, which can be shared across all downstream tasks.

## 2 Background

### 2.1 Matrix factorization

The matrix factorization methods can decompose a large matrix $W \in \mathbb{R}^{M \times N}$ into smaller matrice:

$$W = U\Sigma V^\top \approx (U_r \Sigma_r)V_r^\top = P_r Q_r, \quad (1)$$

where $U \in \mathbb{R}^{M \times l}$, $V \in \mathbb{R}^{N \times l}$, and $l$ is the rank of matrix $W$. $\Sigma$ is a diagonal matrix composed of non-

---

[1]For a weight matrix $W \in \mathbb{R}^{M \times N}$, the computational complexity of SVD is $O(\min(MN^2, M^2N))$

zero singular values $diag(\sigma_1, , ..., \sigma_l)$, where $\sigma_1 \geq \sigma_2 \geq \cdots \sigma_l > 0$. $U_r$, $\Sigma_r$, and $V_r$ are the truncated matrices with rank $r$ and serve as approximations to the original matrix. In practice, it is common to represent matrix $W$ as the product of two smaller matrices $P \in \mathbb{R}^{M \times r}$ and $Q \in \mathbb{R}^{r \times N}$, by setting $P = US$ and $Q = V^T$. Given Equation (1), the forward pass of a linear layer in the Transformer can be rewritten as below:

$$h = Wx + b = PQx + b, \quad (2)$$

where $x$ is the input feature, $b$ is the bias. The above process can be implemented with two smaller liner layers, whose weight matrix is equal to $P$ and $Q$ correspondingly. The number of parameters in the original weight matrix $W$ is $MN$, and the combined number of parameters in $P$ and $Q$ is $Nr + Mr$. Therefore, by utilizing matrix factorization techniques, we can reduce the model size by $NM - (Nr + Mr)$.

### 2.2 Related work

Transfomer-based language models (Waswani et al., 2017; Devlin et al., 2018; Brown et al., 2020; Raffel et al., 2020) have attracted significant attention and, as a result, there is a growing demand to reduce their model size. Although matrix decomposition approaches, such as LoRA-like models (Hu et al., 2021; Zhang et al., 2023), have demonstrated their effectiveness in adapting pre-trained models to downstream tasks. The predominant focus of previous techniques for compressing language models has been on knowledge distillation (Sanh et al., 2019; Jiao et al., 2019), quantization (Shen et al., 2020; Kim et al., 2021), and pruning (Sanh et al., 2020; Xia et al., 2022). Only a few attempts have been made in the direction of matrix decomposition towards the goal of compressing language model.

The closest prior research direction to this paper involves approaches that compress Transformer-based language models using matrix factorization: FWSVD (Hsu et al., 2022) and TFWSVD (Hua et al., 2022). However, there exist at least two inherent drawbacks in these methods that can potentially cause performance bottlenecks. Firstly, the importance of parameters is assumed to remain static throughout the training process. Secondly, a uniform rank budget is typically allocated to all weight matrices. In contrast to these approaches, our proposed RankDyna is capable of addressing both challenges.

Another research direction related to our work is the importance criteria for parameters. Using the magnitudes of parameters as a measure of their importance is a straightforward and intuitive approach (Zhu and Gupta, 2017; Renda et al., 2020). However, this basic metric fails to quantify the actual contributions of parameters toward the overall performance. A more reliable measurement is sensitivity, which approximates the change in the loss function resulting from the removal of a specific parameter (Molchanov et al., 2019; Sanh et al., 2020), which is also the measurement adopted by proposed RankDyna.

## 3 Methodology

In this section, we introduce RankDyna, a solution that enables dynamic resource allocation by monitoring changes in parameter importance during fine-tuning. This approach effectively addresses the above mentioned issues in previous work. As an overview, we outline our proposed RankDyna in Algorithm 1.

### 3.1 Problem statement

RankDyna aims to compress a pre-trained language model $\mathcal{M}$ by decomposing the weight matrices set $\mathcal{W}$ in its linear layers [2]. At the initial step (denoted as step 0), the total number of ranks $R^{(0)}$ in all matrices of model $\mathcal{M}$ is computed as:

$$R^{(0)} = \sum_{W \in \mathcal{W}} l_W, \qquad (3)$$

where $l_W$ is the original rank number of matrix $W$. Given the desired compression ratio $c$, the targeted number of overall ranks $R^{(T)}$ after $T$ steps of fine-tuning is determined as follows:

$$R^{(T)} = \left\lceil c \cdot R^{(0)} \right\rceil, \qquad (4)$$

where $\lceil \cdot \rceil$ is the rounding up operator. Given a targeted rank budget $R^{(T)}$, our task is to search for the optimal resource allocation for model $\mathcal{M}$.

### 3.2 Singular group

Besides Equation (1), an alternative perspective on matrix decomposition is to approximate $W \in$

---

[2]For Transformer-based models, $\mathcal{W}$ includes various types of weight matrices, such as $W_q$ (query), $W_k$ (key), $W_v$ (value), $W_o$ (attention output), $W_{f_1}$ (first FFN), and $W_{f_2}$ (second FFN).

$\mathbb{R}^{M \times N}$ as the sum of the singular groups:

$$W = \sum_{k=1}^{l} \sigma_k u_k v_k^\top \approx \sum_{k=1}^{r} (\sigma_k u_k) v_k^\top = \sum_{k=1}^{r} (p_k) q_k^\top,$$
$$(5)$$

where $\sigma_k$ is the $k$-th singular value. $u_k$ and $v_k$ are $k$-th columns in $U$ and $V$. $p_k$ and $q_k$ are $k$-th columns in $P$ and $Q$. The approximation occurs when the target rank $r$ is less than the original rank $l$. Note that, setting the $k$-th singular value $\sigma_k$ to zero will completely eliminate the effect of the singular group $\phi_k = \{u_k, \sigma_k, v_k^\top\}$ in the reconstruction process. Based on this observation, we adopt singular group $\phi_k$ as the fundamental unit for reconstructing the matrix $W$.

---

**Algorithm 1:** RankDyna

**Input:** Data set $\mathcal{D}$, total training steps $T$, targeted compress ratio $c$.

1 **for** $t = 1, ..., T$ **do**
2    Sample a mini-batch from $\mathcal{D}$;
3    Calculate current rank $R^{(t)}$ through Equation (12);
4    **for** *each* $\Phi(W)^{(t)}$ *in* $\Phi(\mathcal{M})^{(t)}$ **do**
5      **for** *each* $\phi_k$ *in* $\Phi(W)^{(t)}$ **do**
6        Update gradients through Equation (6) – (8);
7        Compute importance score $\mathcal{I}(\phi_k^{(t)})$ in Equation (9);
8        Update moving average score $\bar{\mathcal{I}}(\phi_k^{(t)})$ in Equation (11);
9    Update global remained set $\Phi(\mathcal{M})^{(t)}$ and all the local remained set $\Phi(W)^{(t)}$;
10 **for** *each weight matrix $W$ in Model $\mathcal{M}$* **do**
11    Compute $P$ and $Q$ by singular groups in $\Phi(W)^{(T)}$ through Equation (5);

**Output:** Compressed model $\widehat{\mathcal{M}}$

---

### 3.3 Iterative approximation

In this part, we discuss how to update singular groups during the training. To start with, we initialize $U$, $\Sigma$, and $V$ by performing a standard Singular Value Decomposition (SVD) on the weight matrix $W$. At the $t$-th step, a stochastic gradient descent step is taken to update elements in each singular group. For $\Sigma_k$, the update of its only non-zero element $\sigma_k$ at step $t$ can be described as follows:

$$\sigma_k^{(t)} \leftarrow \sigma_k^{(t-1)} - \eta \nabla_{\sigma_k} \mathcal{L}, \qquad (6)$$

where $\eta$ is the learning rate. For the $i$-th parameter in vector $u_k$ or $v_k$, the update at step $t$ can be denoted as:

$$u_{k,i}^{(t)} \leftarrow u_{k,i}^{(t-1)} - \eta \nabla_{u_{k,i}} \mathcal{L}. \qquad (7)$$

$$v_{k,i}^{(t)} \leftarrow v_{k,i}^{(t-1)} - \eta \nabla_{v_{k,i}} \mathcal{L}. \qquad (8)$$

### 3.4 Importance score of singular-group

Unlike previous work that evaluates the importance of individual parameters, our proposed RankDyna estimates the importance changes for each singular group as a whole. Specifically, the importance score of a singular group is defined as follows:

$$\mathcal{I}(\phi_k^{(t)}) = \mathcal{S}(\sigma_k^{(t)}) + \sum_{i=1}^{d_1} \mathcal{S}(u_{k,i}^{(t)}) + \sum_{j=1}^{d_2} \mathcal{S}(v_{k,j}^{(t)}), \qquad (9)$$

where $\mathcal{S}(\cdot)$ is a score function to measure the change in loss if we remove a certain parameter, $d_1$ and $d_2$ represent the dimension of $u_k$ and $v_k$ respectively. Given any parameter $\theta$, the change in loss of removing $\theta$ can be formulated by the second-order Taylor series expansion (Le Cun et al., 1989):

$$\begin{aligned} \mathcal{S}(\theta) &= |\Delta \mathcal{L}_\theta| = |\mathcal{L} - \mathcal{L}_{\neg\theta}| \\ &= |\theta^\top \nabla_\theta \mathcal{L} + \frac{1}{2}\theta^\top H \theta + \mathcal{O}(||\theta||^3)|, \end{aligned} \qquad (10)$$

where $\mathcal{L}_{\neg\theta}$ is the loss with parameter $\theta$ zeroed out, and $H$ is the Hessian matrix. The larger the value of $\mathcal{S}(\theta)$, the greater impact the parameter $\theta$ will have on the loss $\mathcal{L}$, indicating the higher importance. Note that, the importance comparisons based on $\mathcal{S}(\theta)$ are not limited to parameters within the same matrix or layer, but are applicable to any parameters in model $\mathcal{M}$. More details about the calculation can be found in Section 4.5.2.

Previous work (Zhang et al., 2022) has pointed out that: the importance scores calculated using Equation (10) may exhibit fluctuations, as they are computed on randomly sampled mini-batches. In practice, these fluctuations can be mitigated by applying the "momentum" scheme (Rumelhart et al., 1986), where the next update is determined as a weighted average of the current update and the previous score:

$$\bar{\mathcal{I}}(\phi_k^{(t)}) = \beta \cdot \bar{\mathcal{I}}(\phi_k^{(t-1)}) + (1 - \beta) \cdot \mathcal{I}(\phi_k^{(t)}) \qquad (11)$$

### 3.5 Rank allocation scheme

The approach for tracking changes in parameter importance has been introduced in the previous sections. In this part, we will discuss how to integrate it with dynamic rank resource allocation.

**Scheduler for adjusting rank budget**. Inspired by the sparsity scheduler widely used in pruning (Zhu and Gupta, 2017; Sanh et al., 2020), we gradually decrease the rank budget during training by a dynamic scheduler. Specifically for our task, the scheduler works as three stages. 1) Warm-up phase. The initial $t_0$ steps are considered the warm-up phase, during which the current rank budget is kept as $R^{(0)}$ in Equation (3). 2) Cool-down phase. The last $t_1$ steps are treated as the cool-down phase, where the targeted rank budget $R^{(T)}$ computed in Equation (4) has already been attained and is maintained till the end of training. 3) Adjusting phase. The steps between $t_0$ and $t_1$ are adjusting phase, where the current $R^{(t)}$ decreases as follows:

$$R^{(t)} = R^{(T)} + \left\lceil (R^{(0)} - R^{(T)})(1 - \frac{t - t_0 - t_1}{T - t_0 - t_1})^3 \right\rceil \quad (12)$$

**Global rank allocation**. To evaluate the importance differences among parameters across different weight matrices, we maintain two sets of singular groups: 1) The local remaining set $\Phi(W)$, which comprises all singular groups of weight matrix $W$; and 2) The global remaining set, which is the union of all local remaining sets, denoted as $\Phi(\mathcal{M})$. Initially, remaining set $\Phi(\mathcal{M})^{(0)}$ contains all singular groups in model $\mathcal{M}$, thus $|\Phi(\mathcal{M})^{(0)}|$ is equal to $R^{(0)}$. At step $t$, for any singular group $\phi$, if its importance score $\mathcal{I}(\phi^{(t)})$ is not among the top-$R^{(t)}$ of $\Phi(\mathcal{M})^{(t)}$, RankDyna will remove $\phi$ from both global remained set $\Phi(\mathcal{M})^{(t)}$ and local remained set $\Phi(W)^{(t)}$.

## 4 Experiment

### 4.1 Tasks and datasets

RankDyna and baselines are evaluated on GLUE benchmark (Wang et al., 2019) and the NER task on the CoNLL-2003 dataset (Sang and De Meulder, 2003). Also, we reported the results of the summarization task (ROUGE score) on SAMSUM dataset, and language modeling task (Perplexity) on the Wikitext2 and PTB datasets. More details can be found in Appendix A.1.

### 4.2 Implementation details and baselines

We utilize pre-trained BERT (Devlin et al., 2018), GPT-2 (Radford et al., 2019), and BART (Lewis

Table 1: Results of CoNLL and GLUE benchmark. All model sizes reported here exclude the embedding layer. G-Avg represents the average of GLUE tasks, while A-Avg denotes the average of all tasks, including CoNLL.

| Task | #Param | CoNLL | CoLA | MNLI | MRPC | QNLI | QQP | SST-2 | STS-B | G-Avg | A-Avg |
|------|--------|-------|------|------|------|------|-----|-------|-------|-------|-------|
| Bert_base | 86.2M | 94.1 | 56.2 | 84.7 | 87.4 | 91.3 | 87.8 | 93 | 88.5 | 84.1 | 85.4 |
| SVD | 43.1M | 92.4 | 40.5 | 82.8 | 84.1 | 89.6 | 87.3 | 90.9 | 85.7 | 80.1 | 81.6 |
| FWSVD | 43.1M | 93.2 | 49.4 | 83.0 | 88.0 | 89.5 | 87.6 | **91.2** | 87.0 | 82.2 | 83.6 |
| TFWSVD | 43.1M | 94.2 | 52.2 | **83.4** | 89.0 | 90.3 | 86.9 | 91.1 | 88.5 | 83.1 | 84.4 |
| RankDyna | 43.1M | **94.3** | **54.5** | 83.2 | **89.5** | **90.6** | **88.0** | 90.9 | **89.0** | **83.7** | **85.0** |
| SVD | 26.6M | 92.8 | 19.3 | 81.0 | 82.0 | 86.6 | 86.9 | 89.2 | 80.6 | 75.1 | 77.3 |
| FWSVD | 26.6M | 92.9 | 38.7 | 81.4 | 80.3 | 88.0 | 87.2 | 88.4 | 82.9 | 78.1 | 80.0 |
| TFWSVD | 26.6M | **93.5** | 39.3 | 82.2 | **88.3** | 88.8 | 87.0 | 89.9 | 87.0 | 80.4 | 82.0 |
| RankDyna | 26.6M | 93.3 | **49.1** | **82.5** | 88.0 | **89.2** | 87.5 | 90.7 | 88.3 | **82.1** | **83.6** |
| SVD | 13.9M | 90.4 | 13.8 | 78.0 | 82.0 | 79.6 | 84.1 | 87.5 | 58.7 | 69.1 | 71.7 |
| FWSVD | 13.9M | 3.5 | 18.7 | 78.2 | 78.6 | 82.3 | 84.5 | 88.9 | 67.9 | 71.3 | 62.8 |
| TFWSVD | 13.9M | **91.9** | 21.4 | 79.1 | **85.0** | 84.3 | 85.9 | **89.0** | 86.0 | 75.8 | 77.8 |
| RankDyna | 13.9M | 91.6 | **36.2** | **81.3** | 84.7 | **87.6** | 86.8 | 88.2 | **86.4** | **78.7** | **80.4** |
| SVD | 6.4M | 87.4 | 0.0 | 71.6 | 74.7 | 65.8 | 79.8 | 84.3 | 22.7 | 57.0 | 60.8 |
| FWSVD | 6.4M | **88.4** | 17.2 | 73.5 | 80.9 | 71.2 | 81.3 | 84.9 | 39.2 | 64.0 | 67.1 |
| TFWSVD | 6.4M | 87.8 | 17.8 | 76.7 | 81.2 | 76.5 | 83.4 | 81.3 | 45.3 | 66.0 | 68.8 |
| RankDyna | 6.4M | **88.4** | **21.3** | **79.2** | **82.1** | **85.5** | **86.1** | **87.5** | **83.3** | **75.0** | **76.7** |

et al., 2020) as the starting points for all models, namely RankDyna, TFWSVD, FWSVD, and SVD. For TFWSVD, FWSVD, and SVD, we first fine-tune the base model using task-specific data for 8 epochs and then apply these low-rank factorization methods to the obtained task-specific models. Finally, we conducted an additional round of fine-tuning with 3 epochs. For RankDyna, we directly fine-tune the base model for 8 epochs on each downstream task. More details can be found in Appendix A.2.

## 4.3 Performance comparisons with SOTA

The results of GLUE tasks and one NER task (CoNLL) are presented in Table 1. Our RankDyna model, with 43.1M linear parameters, achieves a G-Avg score of 83.7 and an A-Avg score of 85.0. In fact, RankDyna not only surpasses the scores of state-of-the-art (SOTA) compression methods (SVD, FWVSD, TFWSVD) that require additional fine-tuning, but also even beats the full-size Bert base model in some tasks (e.g., QQP, STSB, MRPC, and CoNLL) with half parameter budget. This suggests that our RankDyna functions as both a compression method and an adaptation approach. This is not surprising, considering the proven effectiveness of low-rank adapters like LoRA in fine-tuning the general pre-trained model for downstream tasks (Hu et al., 2021).

RankDyna consistently yields good results on

| Model | R1 | R2 | RL |
|-------|----|----|----|
| BART-base | 52.1 | 27.3 | 43.5 |
| SVD | 45.3 | 21.3 | 37.5 |
| FWSVD | 47.0 | 22.6 | 38.8 |
| TFWSVD | 47.8 | 23.1 | 39.3 |
| RankDyna | 48.4 | 23.4 | 39.7 |

Table 2: Compressing BART model for summarization task SAMSUM (Higher is better). By applying a desired rank ratio of 0.2 across all methods, we reduce these 94.6M parameters to 37.1M.

all the tasks. Moreover, as the compression rate increases, the advantage of RankDyna becomes more apparent. Even under the limited budget of 6.4M linear parameters, our RankDyna is capable of maintaining a respectable G-Avg score of 75.0 and A-Avg score of 76.7. These scores are comparable to the G-Avg score and A-Avg score generated by other methods operating under a more generous 13.9M parameter budget.

In most settings, the standard SVD method shows the poorest performance. But it exhibits greater stability compared to the two importance-aware decomposition methods. TFWSVD is the second-best performer in most cases. However, this method heavily relies on a numerical optimization process for matrix factorization, which may be trapped in local optima (Hua et al., 2022). However, TFWSVD is the approach that requires the most computation resources. More details about

Figure 1: Importance changes of singular groups. To simulate the starting points of FWSVD and TFWSVD, we further fine-tune the Bert base model on the STSB dataset for 1200 steps. In the initial step, we sort the singular groups within each weight matrix in descending order, based on their importance scores. We then select specific groups (e.g., Group 1, 10, 40, 70 in this case) and monitor their relative importance positions throughout subsequent steps. At step 200, we remove $80\%$ of less important singular groups in $W_q$, $W_k$, $W_v$ and $W_{f_1}$, and observe the changes in importance of the singular groups in $W_o$ and $W_{f_2}$.

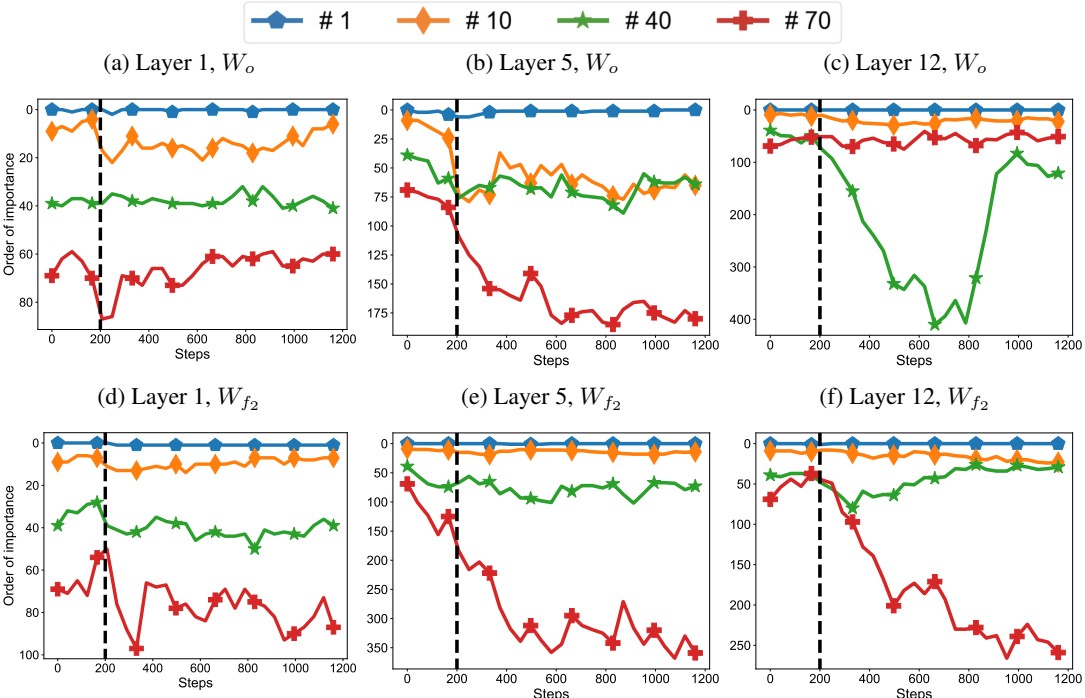

the training time can be found in Appendix A.3.

We have additional results on generation tasks, such as summarization and language modeling. Table 2 shows the performance of compressing BART model on summarization dataset SAMSUM, and Table 3 is the comparison of compressing GPT-2 model through language modeling task on dataset wiki2 and PTB. Generally, these results show the similar patterns we observed in GLUE dataset: our RankDyna can achieve the best performance, TFWSVD is the second best performer, and FWSVD shows consistently better performance than traditional SVD.

### 4.4 Importance changes during fine-tuning

The two underlying assumptions held by previous importance-aware decomposition methods, FWSVD and TFWSVD, are as follows: 1) Stable assumption: The distribution of importance over parameters is supposed to remain relatively stable during the subsequent fine-tuning process. 2) Independent assumption: The matrices are assumed to be independent of each other.

As can be seen from Figure 1, the most impor-

| Model | wiki-2 | PTB |
|---|---|---|
| GPT-2 | 20.9 | 21.4 |
| SVD | 124.7 | 79.6 |
| FWSVD | 121.6 | 72.2 |
| TFWSVD | 79.3 | 62.9 |
| RankDyna | 77.7 | 39.4 |

Table 3: Compressing GPT-2 model for language modeling tasks. (Lower is better) By applying a rank ratio of 0.2 for all methods, we reduced this parameter count to 28.9M.

tant singular groups (such as Group 1 and 10) remain relatively stable throughout the entire process. However, other groups, such as Group 40 and Group 70, already exhibit fluctuation patterns before the removal of singular groups. These observations definitely contrast with the stable assumption held by FWSVD and TFWSVD.

Furthermore, we observed significant fluctuations in all matrices immediately after removing unimportant singular groups in previous layers. This observation suggests that the removal of singular groups in one weight matrix could trigger chain reactions in the importance scores of subsequent

layers, demonstrating the invalidity of the independent assumption held by FWSVD and SVD.

In contrast, the success of RankDyna does not rely on these two assumptions, as we actively monitor the dynamic changes in importance for all singular groups across different layers.

## 4.5 Ablation Study

### 4.5.1 Global vs. local rank allocation

In order to examine the impact of global rank allocation, as discussed in Section 3.5, we introduce a variant baseline approach. This approach selects the top-$R^{(t)}$ singular groups within each matrix, instead of utilizing the global allocation strategy utilized by RankDyna.

Figure 2: Comparison of RankDyna (green line named "global") and its baseline variant (blue line named "local") with STSB task. The red dashed line in Figure 2b represents the rank budget scheduler (Section 3.5).

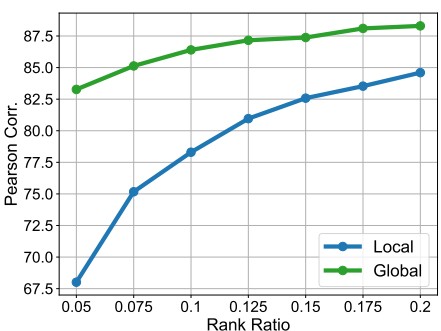

(a) Different rank budget levels.

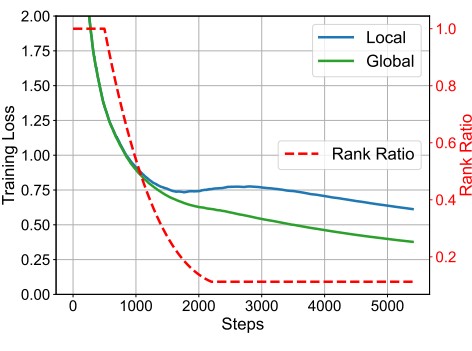

(b) Training loss.

As shown in Figure 2a, the performance of RankDyna (global) consistently surpasses that of its variant baseline (local) across different parameter budget levels. To further analyze the underlying factors behind this phenomenon, we look into the trends of loss changes shown in Figure 2b. Initially, both methods exhibit similar behavior. However, as we dynamically adjust the rank budgets, RankDyna (global) starts to outperform the baseline (local)

by a significant margin. We also observed a slight increase in the loss of the baseline (local) after reducing the parameter budget, indicating that the baseline may mistakenly delete important parameters, resulting in a drop in performance.

### 4.5.2 Approximation of importance score

In this section, we examine different approximations of importance scores shown in Equation (10). To conduct this investigation, we fist fine-tune the base BERT model into task-specific models and then calculate the different scores at the convergence stage. The results shown in Table 4 indicate that the second-order term in Equation (10) and the Fisher information can be considered negligible when compared to the first-order term. Specifically, the magnitude of the first-order term in Equation (10) (around E-05) is significantly larger than that of the second-order term (around E-09) and the Fisher information (around E-08). Therefore, in this work, we use the first-order term to approximate the importance score shown in Equation (10).

Note that the scenario we studied here is the trained task-specific model. FWSVD and TFWSVD assume that the first-order term is zero in such models, which leads to the utilization of Fisher information as their importance score. However, our findings demonstrate that the first-order term remains non-negligible, thereby challenging the validity of solely relying on Fisher information as the importance score.

## 4.6 Rank distribution over layers

Figure 3 shows the rank distributions generated by RankDyna on MNLI with 13.9M and 6.4M linear layer parameter budgets. The rank distributions of the fixed assignment can be found in the Appendix.

RankDyna can effectively assign different ranks to the weight matrices based on their importance toward performance. Taking the 13.9M model as an example, each weight matrix is supposed to be assigned 77 ranks. Figure 3a illustrates that when using RankDyna, the majority of weight matrices are assigned ranks approximately around 77, while only a small number of them attaining an exact rank of 77. As the parameter budget decreases from 13.9M to 6.4M, more deep green blocks (representing matrices with low-rank assignments) appear. However, the most important matrices are still assigned high ranks despite the tighter budget.

We have also observed that RankDyna tends to allocate a greater portion of the budget to the front

Table 4: Comparison of first-order, second-order and empirical Fisher information importance score in trained task-specific models. We fine-tune Bert model on different tasks for 6 epochs and calculate the average of importance scores of the last 100 steps. The first-order and second-order term are the first and second term in Equation (10). The definition and calculation of Fisher information can be found in Appendix.

|  | CoNLL | CoLA | MRPC | QNLI | SST-2 | STSB |
|---|---|---|---|---|---|---|
| First-order term | 2.53E-05 | 2.84E-05 | 2.59E-05 | 2.74E-05 | 2.94E-05 | 2.50E-05 |
| Second-order term | 6.84E-09 | 7.30E-09 | 8.16E-09 | 5.16E-09 | 6.06E-09 | 8.07E-09 |
| Fisher information | 1.23E-08 | 1.17E-08 | 1.03E-08 | 9.83E-09 | 1.11E-08 | 1.03E-08 |

Figure 3: The resulting rank of each weight matrix when compressing Bert-base on MNLI with RankDyna. The x-axis represents the layer index, while the y-axis represents different types of weight matrices.

(a) 13.9M linear parameters (equal to rank ratio 0.1)      (b) 6.4M linear parameters (equal to rank ratio 0.05)

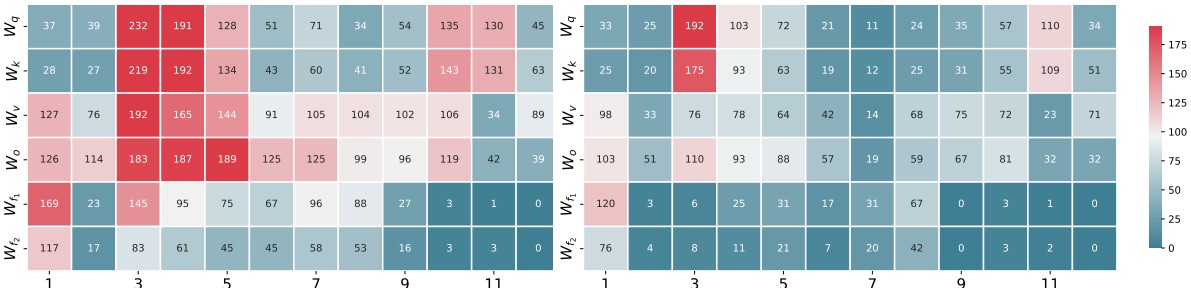

layers. This trend is evident in both Figure 3a and Figure 3b, where a higher number of ranks is preserved in the front layers, particularly in the attention matrices Q, K, and V. In contrast, more redundant parameters are eliminated in the tail layers. For instance, both Figure 3a and Figure 3b demonstrate that the last FFN layers receive zero rank allocations. This behavior aligns with the empirical observation presented in Q-BERT (Shen et al., 2020) for the MNLI task.

## 4.7 Selection strategy

In standard SVD, only the portion associated with large singular values will be kept in the compressed model. It is already established that this selection strategy may not necessarily align with the ultimate objective of maintaining task performance. Then, under the selection strategy of RankDyna, which parameters are chosen as the important ones? Figure 4 illustrates the different selection strategies conducted by SVD and RankDyna. As can be seen from Figure 4a, only the singular-groups with the largest singular values are treated as the important ones under SVD. In contrast, RankDyna will pick up more groups from those with larger singular values, and at the same time, it will also choose certain singular groups with smaller singular values as long as they are important for the final performance. The behavior of RankDyna demonstrates a well-balanced objective that considers both the

Figure 4: Comparison of selection strategy of SVD and RankDyna. Each bar here represents a subset containing 55 singular groups, sorted by the average singular values. The colors denote the portion of singular groups that are selected as the top important groups. The green color represents the selection made by SVD, while the red color indicates the selection made by RankDyna. The matrix here is the Value matrix of the last layer in 26.6M model fine-tuned on QNLI.

(a) SVD                 (b) RankDyna

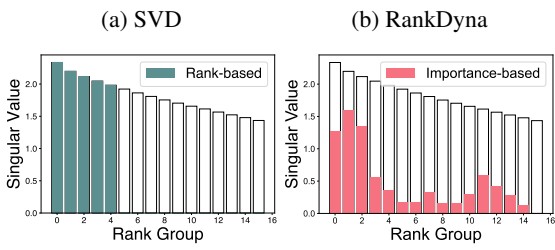

numerical features of the weight matrix itself and the model performance.

## 5 Conclusion

RankDyna provides a comprehensive solution for compressing Transformer-based models. Through the integration of importance tracking and dynamic allocation of ranks across different layers, RankDyna addresses the unsolved issues left by the previous importance-aware decomposition methods. Extensive results demonstrate that RankDyna outperforms current state-of-the-art (SOTA) models by a significant margin in terms of performance.

## Limitations

The primary limitation of RankDyna lies in its potential requirement for additional memory to keep tracking the importance scores during fine-tuning. For each weight matrix, an equivalent amount of memory is needed to enable the computation of importance scores. The need for additional memory is specific to the training phase only. During inference, the compact models produced by RankDyna function similarly to models generated by other matrix decomposition methods.

## 6   Ethical Statement

For our experiments, we used open datasets without sensitive information, which have been widely mentioned in previous work. No licenses are required for the GLUE dataset and CoNLL dataset. In the implementation of our model, we do not think there is an obvious issue that may lead to a risk to ethics.

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

## A Appendix

### A.1 Datasets

Two single-sentence tasks are included in our evaluation: CoLA (Warstadt et al., 2018) measured in Matthew's correlation, SST2 (Socher et al., 2013) measured in classification accuracy. Three sentence similarity tasks are considered: MRPC (Dolan et al., 2005) measured in F-1 score, STS-B (Cer et al., 2017) measured in Pearson-Spearman correlation, QQP (Chen et al., 2018) measured in F-1 score. Also, three natural language inference tasks are conducted: MNLI (Williams et al., 2018) measured in classification accuracy with the average of the matched and mismatched subsets, QNLI (Rajpurkar et al., 2016) measured in accuracy.

SAMSUM dataset contains staged chat conversations and corresponding summaries made by linguists (Gliwa et al., 2019), including 14,732 training examples and 819 test examples. The performance is evaluated through Rounded ROUGE values (R1/R2/RL). We evaluate language modeling task through metric perplexity on two popular model datasets: PTB (Marcus et al., 1993) and WikiText-2 (Merity et al., 2016).

### A.2 Implementation details

The BERT model in this paper has 110 M parameters, out of which 94.6M correspond to the parameters in the linear layers. The BART model we utilize comprises 133M parameters, out of which 94.6M correspond to the parameters in the linear layers. The GPT-2 model we use has 117M parameters, and 81.0M parameters are associated with its linear (Conv1D) layers.

Only the linear layers within the transformer blocks are compressed in this study, and non-Transformer modules, such as the token embedding, remain uncompressed. The settings not explicitly mentioned in this work utilize the default configurations of the HuggingFace Transformer library (Wolf et al., 2020).

### A.3 Training Time

RankDyna begins with a generic pre-trained model, which can be reused across all downstream tasks. In contrast, the other factorization methods mentioned in the paper (SVD, FWSVD, TFWSVD) require a task-specific fine-tuned model for factorization. Consequently, as the model size increases and factorization becomes more computationally expensive, RankDyna saves more computational

Table 5: Hyper-parameter setting for training Rank-Dyna.

| Dataset | Batch Size | Initial | Final |
|---------|:----------:|:-------:|:-----:|
| **CoNLL** | 16 | 500 | 2500 |
| **CoLA** | 32 | 500 | 1500 |
| **MNLI** | 32 | 5400 | 27000 |
| **MRPC** | 8 | 500 | 1500 |
| **QNLI** | 32 | 2000 | 12000 |
| **QQP** | 32 | 5400 | 27000 |
| **SST-2** | 16 | 500 | 1500 |
| **STSB** | 16 | 500 | 2500 |

resources by only requiring one base model that can be applied to all tasks. Furthermore, unlike the other methods that require two rounds of model fine-tuning (one before compression and one after), RankDyna accomplishes compression with only one round of fine-tuning.

In comparison to SVD, FWSVD requires additional time to compute Fisher information for each specific task. TFWSVD requires even more time as it performs iterative calculations of decomposition.

### A.4 Fisher information

Fisher information measures the amount of information that a given dataset $D$ provides about a model parameter $w$. However, obtaining precise values for Fisher information is generally intractable due to the computational complexity. In practice, the empirical Fisher information is estimated as follows:

$$
\begin{aligned}
I_w &= E\left[\left(\frac{\partial}{\partial w}\log p(\mathbf{D}|\mathbf{w})\right)^2\right] \\
&\approx \frac{1}{|\mathbf{D}|}\sum_{i=1}^{|\mathbf{D}|}\left(\frac{\partial}{\partial w}(d_i; w)\right)^2 = \hat{I}_w.
\end{aligned}
\tag{13}
$$

Given a target task objective (e.g., cross-entropy for a classification task), the estimated information $\hat{I}_w$ accumulates the squared gradients over the training data $d_i \in \mathcal{D}$. The parameters that cause large absolute gradient of the task objective will have a large value in $\hat{I}_w$, and are considered important to the target task.

## B Training Details

We use a fixed $\beta = 0.85$ for importance score momentum calculation in Equation (11), and learning rate 1E-04 throughout all experiments. We list additional hyper-parameter settings used by RankDyna in Table 5.

Figure 5: In Section 4.6, we presented the rank distribution of RankDyna on the MNLI dataset. Here we provide a demonstration of the rank allocation when assigning a uniform rank to each weight matrix. The rank distribution on MNLI with the fixed assignment. The x-axis represents the layer index, while the y-axis represents different types of weight matrices.

(a) 13.9M linear parameters (equal to rank ratio 0.1)     (b) 6.4M linear parameters (equal to rank ratio 0.05)

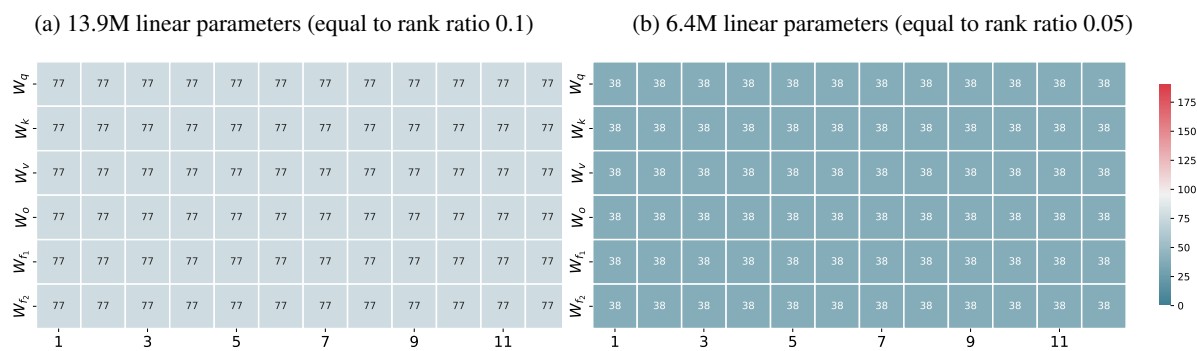

Figure 6: The resulting rank of each weight matrix when compressing Bert-base on **QNLI** with RankDyna. The x-axis represents the layer index, while the y-axis represents different types of weight matrices.

(a) 13.9M linear parameters (equal to rank ratio 0.1)     (b) 6.4M linear parameters (equal to rank ratio 0.05)

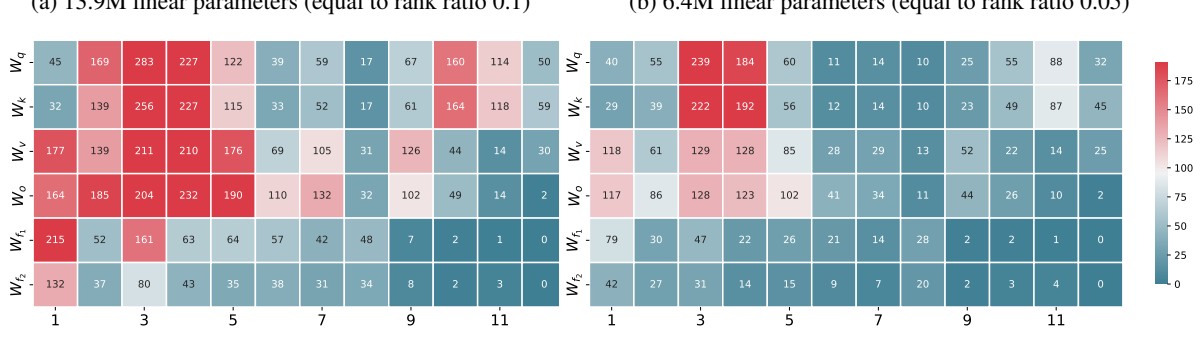

Figure 7: The resulting rank of each weight matrix when compressing Bert-base on **QQP** with RankDyna. The x-axis represents the layer index, while the y-axis represents different types of weight matrices.

(a) 13.9M linear parameters (equal to rank ratio 0.1)     (b) 6.4M linear parameters (equal to rank ratio 0.05)

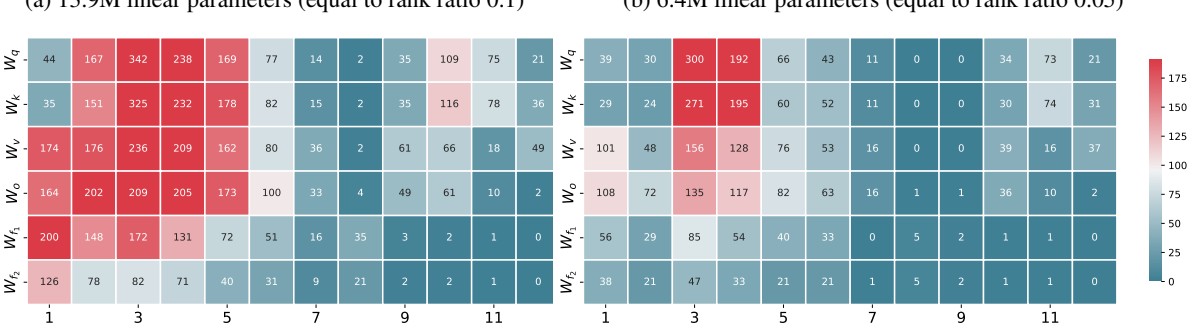

Figure 8: The resulting rank of each weight matrix when compressing Bert-base on **STSB** with RankDyna. The x-axis represents the layer index, while the y-axis represents different types of weight matrices.

(a) 13.9M linear parameters (equal to rank ratio 0.1)  (b) 6.4M linear parameters (equal to rank ratio 0.05)

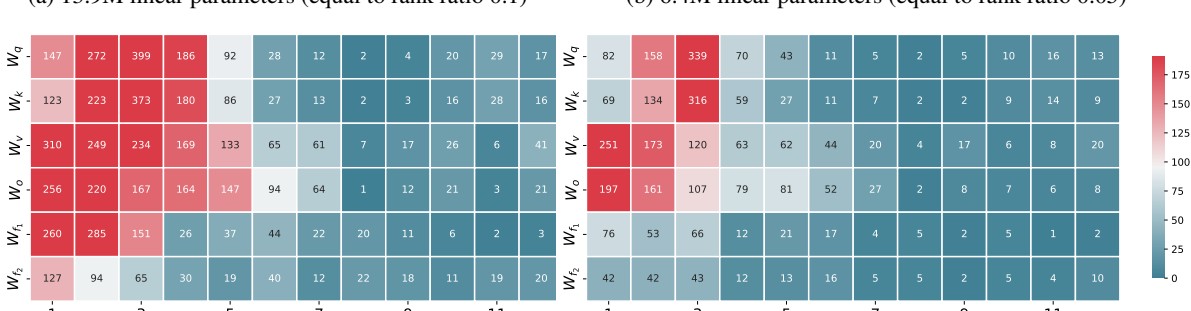