# OpenReview forum: "Dynamic Low-rank Estimation for Transformer-based Language Models"
_EMNLP/2023/Conference — EMNLP 2023 Findings_

### Official Review · Reviewer_YbZE · 2023-07-20

**Soundness:** 3

**Excitement:**

3: Ambivalent: It has merits (e.g., it reports state-of-the-art results, the idea is nice), but there are key weaknesses (e.g., it describes incremental work), and it can significantly benefit from another round of revision. However, I won't object to accepting it if my co-reviewers champion it.

**Paper Topic And Main Contributions:**

This paper proposed RankDyna, a matrix decomposition method  that enables dynamic rank resource allocation among matrices across different layers during the training process.

**Reasons To Accept:**

(1) The motivation and the design of proposed method seem reasonable.

(2)  Organized sections with clear statement in mathematical fomulation.

(3)  The improvement is significant.

**Reasons To Reject:**

(1) The proposed method will introduce extra computation and memory cost. However, there is no efficiency analysis, such as  detailed training time (even though it is mentioned briefly in appendix) and memory usage comparison.

(2)  Using only bert_base (encoder-only architecture of transformer-base models) for experiment makes this paper a little bit overclaim.   Also, it is better to verify effiectiveness of  RankDyna on NLG tasks.

(3)  The idea is not that novel since  there are already works such as FWSVD and TFWSVD. The main contribution of this paper is to introduce dynamic estimation compared to preivous works.

**Reproducibility:**

4: Could mostly reproduce the results, but there may be some variation because of sample variance or minor variations in their interpretation of the protocol or method.

**Reviewer Confidence:**

3: Pretty sure, but there's a chance I missed something. Although I have a good feel for this area in general, I did not carefully check the paper's details, e.g., the math, experimental design, or novelty.

---

> ### Author Rebuttal · Authors · 2023-08-29
>
> Please refer to response to Reviewer **WKgh** for additional results with GPT-2 and BART.  We answer all other questions as follows.
>
> > Question 1. The proposed method will introduce extra computation and memory cost. However, there is no efficiency analysis, such as detailed training time (even though it is mentioned briefly in appendix) and memory usage comparison.
>
> Taking the PTB task (GPT-2) for example, the training time of different models are as follows. The time unit is minutes. As we mentioned in Question 1 to reviewer **NNuk**, SVD/WSVD/TFWSVD will require two-rounds of finetuning, resulting in a larger total number of training epochs compared to our RankDyna. For WSVD and TFWSVD, additional time is needed for calculating the Fisher information or numerical estimation required for matrix factorization. Our RankDyna indeed costs more time per epoch as we will do more calculations. Nonetheless, since we only require one round of fine-tuning, our total training time is the shortest among all methods.
> |          | Time per Epoch | Training Epoch | Basic SVD| Fisher Information | Numerical Estimation | Overall Training time |   |   |   |
> |----------|----------------------|----------------|---------------|-------------------------|---------------------------|-------------------------------|---|---|---|
> | SVD      | 1.9                  | 6              | 0.22          | -                       | -                         | 11.6                       |   |   |   |
> | WSVD     | 1.9                  | 6              | 0.22          | 6.3                     | -                         | 17.9                       |   |   |   |
> | TFWSVD   | 1.9                  | 6              | 0.22          | 6.3                     | 53.2                      | 71.1                       |   |   |   |
> | RankDyna | 2.9                  | 3              | 0.22          | -                       | -                         | 8.9                        |   |   |   |
>
>
> As mentioned in the Limitations section, the primary downside of RankDyna is the need for additional running memory (almost equivalent in size to the model itself). This drawback doesn't cause notable issues for models like GPT-2, BART, or BERT. However, to be honest, when dealing with huge models like GPT-3 or LLAMA, it could potentially lead to memory errors. A possible strategy is to maintain the importance data structures on the CPU, which would help reduce GPU memory consumption.
>
> > Concern 1. The idea is not that novel since there are already works such as FWSVD and TFWSVD. The main contribution of this paper is to introduce dynamic estimation compared to preivous works.
>
> FWSVD/TFWSVD learn importance in a static manner, while our dynamic methodology is not only innovative but also indispensable. As mentioned in Section 4.4, both FWSVD and TFWSVD are built upon two assumptions: the stability assumption and the Independent assumption (line 406 - line 412). These unrealistic assumptions consequently limit their performance. In contrast, our RankDyna operates independently of these two assumptions, leading to better performance outcomes.
>
> Thank you once again for the inspiring suggestion. We will add the above discussion to the final version if the paper is accepted.

---

### Official Review · Reviewer_WKgh · 2023-08-05

**Soundness:** 3

**Excitement:**

3: Ambivalent: It has merits (e.g., it reports state-of-the-art results, the idea is nice), but there are key weaknesses (e.g., it describes incremental work), and it can significantly benefit from another round of revision. However, I won't object to accepting it if my co-reviewers champion it.

**Paper Topic And Main Contributions:**

The authors propose RankDyna, a matrix decomposition method that enables dynamic rank resource allocation among matrices across different layers during the training process. Their proposed method achieves compression and adapts to different downstream task by fine-tuning the model just once.


**Questions For The Authors:**

Did the authors experiment with any of the other transformer based models except BERT?

**Reasons To Accept:**


+ Proposes a matrix decomposition approach that captures dynamic changes the importance associated with singular groups.
+ Extensive experiments demonstrating the importance of the method
+ Reduced requirement of computational and storage resources.

**Reasons To Reject:**

- Demonstrate the results only on the BERT model.

**Reproducibility:**

3: Could reproduce the results with some difficulty. The settings of parameters are underspecified or subjectively determined; the training/evaluation data are not widely available.

**Reviewer Confidence:**

2: Willing to defend my evaluation, but it is fairly likely that I missed some details, didn't understand some central points, or can't be sure about the novelty of the work.

---

> ### Author Rebuttal · Authors · 2023-08-29
>
> We have additional results on seq2seq model BART (summarization task SAMSUM) and GPT-2 (language modeling task wiki2 and PTB). Generally, these results show the similar patterns we observed in GLUE dataset.
>  **Summarization**       | **R1**   | **R2**   | **RL**   |
> |--------------|------|------|------|
> | BART-base| 52.1 | 27.3 | 43.5 |
> | SVD          | 45.3 | 21.3 | 37.5 |
> | WSVD         | 47.0 | 22.6 | 38.8 |
> | TFWSVD       | 47.8 | 23.1 | 39.3  |
> | RankDyna | 48.4 | 23.4 | 39.7  |
>
> The BART model we utilize comprises 133M parameters, out of which 94.6M correspond to the parameters in the linear layers. By applying a desired rank ratio of 0.2 across all methods, we reduce these 94.6M parameters to 37.1M.
>  **LM**   | **wiki-2** | **PTB** |
> ------------|------------|---------|
>  GPT-2-base | 20.9       | 21.4    |
>  SVD        | 124.7      | 79.6    |
>  WSVD       | 121.6      | 72.2    |
>  TFWSVD     | 79.3       | 62.9    |
>  RankDyna   | 77.7       | 39.4    |
>
> Similarly, the GPT-2 model we use has 117M parameters,  and  81.0M parameters are associated with its linear (Conv1D) layers. Through applying a rank ratio of 0.2 for all methods, we reduce this parameter count to 28.9M.
>
> It's worth noting that applying a rank ratio of 0.2 yields a compact model with approximately 35~40% of its original model size. For more details, please refer to the reply to Question 2 for reviewer ****ncwh**** and Section 2.1 in our paper.
>
> Thank you once more for suggesting the inclusion of additional experiments. This is indeed crucial for enhancing the quality of our paper.

---

### Official Review · Reviewer_ncwh · 2023-08-11

**Typos Grammar Style And Presentation Improvements:** 1. It seems the training loss line in…
**Soundness:** 3

**Excitement:**

3: Ambivalent: It has merits (e.g., it reports state-of-the-art results, the idea is nice), but there are key weaknesses (e.g., it describes incremental work), and it can significantly benefit from another round of revision. However, I won't object to accepting it if my co-reviewers champion it.

**Missing References:**

None.

**Paper Topic And Main Contributions:**

This article discusses the topic of parameter compression of Transformer-based language models via matrix decomposition.

They proposed RankDyna, an algorithm that dynamically tracks parameter importance and adjusts the rank budget to obtain a compressed model during fine-tuning.

Their proposed method addresses two dependent assumptions of previous work:
1. The importance distribution of the parameters is invariant during the finetune process.
2. The weights of the different layers are independent of each other.

Their approach achieves reasonable improvements in language model benchmark at different compression rates.

**Questions For The Authors:**

Question A. In Table 2, the results are the average of the last 100 steps and the model is fine-tuned for 6 epochs. While the other figures display the results of the early thousands of steps. Were the findings in Table 2 also consistent in the early days of fine-tuning?

Question B. In line 313, what is the exact number of "top-R"? Like 50%?

Question C. Line 205, why don't these linear layers contain the embedding matrix?

Question D. Doesn't keep track of the importance of parameters during training take extra time? Why?

**Reasons To Accept:**

1. Reasonable idea, decoupled from the assumptions required by previous work.
2. The main results and analytical experiments corroborate the method's effectiveness and the proposed claims.
3. SOTA results.

**Reasons To Reject:**

1. There is only a 12-layer BERT model as the base model and no experiments with generative models.

**Reproducibility:**

4: Could mostly reproduce the results, but there may be some variation because of sample variance or minor variations in their interpretation of the protocol or method.

**Reviewer Confidence:**

3: Pretty sure, but there's a chance I missed something. Although I have a good feel for this area in general, I did not carefully check the paper's details, e.g., the math, experimental design, or novelty.

---

> ### Author Rebuttal · Authors · 2023-08-29
>
> Please refer to response to Reviewer **WKgh** for additional results with GPT-2 and BART. We answer all other questions as follows.
>
> >Question 1. In Table 2, the results are the average of the last 100 steps and the model is fine-tuned for 6 epochs. While the other figures display the results of the early thousands of steps. Were the findings in Table 2 also consistent in the early days of fine-tuning?
>
> FWSVD and TFWVSD hold an assumption that the first-order term of importance score is zero in trained models. Both Table 2 and Figure 1 demonstrate that this assumption is untrue. Note that, the scenarios in  Table 2 and Figure 1 are actually very close. Figure 1 is a converged task-specific model plus 1200 steps, while Table 2 shows 200 steps before obtaining the fine-tuned task-specific model. To maintain consistency and minimize confusion, we will adjust Table 2 to align with the scenario presented in Figure 1 (a task-specific model with an additional 1,200 steps).
>
> Besides clarifying the confusion above, we guess you might be interested in observing the scores during the early stages of fine-tuning.  We show the scores of first 100 steps during the initial fine-tuning. The overall pattern is similar to that of Table 2, with slightly higher values. This is due to the fact that the numbers in Table 2 are closer to the converged states.
>
>  **Scores**         | **CoLA** | **MRPC** | **QNLI** | **SST-2** | **STSB** |
> --------------------|----------|----------|----------|-----------|----------|
>  First-order term   | 2.72e-05 | 2.65e-05 | 3.01e-05 | 2.68e-05  | 2.77e-05 |
>  Second-order term  | 8.66e-09 | 9.02e-09 | 8.34e-09 | 6.05e-09  | 8.87e-09 |
>  Fisher information | 1.08e-08 | 1.05e-08 | 1.07e-08 | 1.03e-08  | 1.04e-08 |
>
> >Question 2. In line 313, what is the exact number of "top-R"? Like 50%?
>
> top-\\( R^{(t)}\\) will change along with the training time \\( t\\) . The final top-\\( R^{(t)}\\) is related to the desired rank number. Taking a square matrix (\\( N \times N\\)) for example, if the targeted compression rate is 50% of its original size, then the rank number will be  \\(  \frac{1}{4} N\\). More details  can be found in Section 2.1.  Similarly, if we wanted to compress the linear layers in the model to around 50% of their original parameters, then the final number of top-\\( R^{(t)}\\) will be around 25%.
>
> >Question 3. Line 205, why don't these linear layers contain the embedding matrix?
>
> We follow the configuration of prior studies such as WSVD/FWSVD, which retains the embedding layer uncompressed. Earlier research [Q-BERT] has indicated that the embedding layer is more sensitive than the regular linear layer. But technically, our method can be applied to the embedding layer as well.
>
> [Q-BERT] Shen, Sheng, Zhen Dong, Jiayu Ye, Linjian Ma, Zhewei Yao, Amir Gholami, Michael W. Mahoney, and Kurt Keutzer. "Q-bert: Hessian based ultra low precision quantization of bert." In *Proceedings of the AAAI Conference on Artificial Intelligence*, vol. 34, no. 05, pp. 8815-8821. 2020.
>
> > Question 4. Doesn't keep track of the importance of parameters during training take extra time? Why?
>
> Indeed, such an operation does introduce some additional time. For detailed information about the training time, please refer to the response provided to Question 1 of reviewer **YbZE**.
>
> > Additional comment: Nowhere is there a definition of what FWSVD and TFWSVD are and their reference.
>
> We mentioned the two previous work several times in the introduction (line 56, line 61-64) and related work (line 169-line172), but it is true that we didn’t denote  FWSVD and TFWSVD clearly. Thank you for bringing this to our attention, and we will make sure to include references to FWSVD and TFWSVD.
>
>  We will add the above discussion to the final version. Thanks again for all the insightful questions.

---

### Official Review · Reviewer_NNuk · 2023-08-15

**Soundness:** 4

**Excitement:**

2: Mediocre: This paper makes marginal contributions (vs non-contemporaneous work), so I would rather not see it in the conference.

**Missing References:**

Mentioned above in reasons to reject.

**Paper Topic And Main Contributions:**

This work proposes a dynamic SVD technique for compressing transformer based language models for downstream tasks. They perform the compression at the granularity of a singular group instead of the level of a parameter. As with other recent methods, they compute an importance score to guide their compression. They only need to fine-tune once instead of twice like most methods.

**Questions For The Authors:**

1. In the paper you mention "RankDyna only requires an one-time factorization on the base model, which can be shared across all downstream tasks." Don't you need to apply RankDyna for every downstream task since you would be working on a separate dataset each time?


**Reasons To Accept:**

1. Simple algorithm which operates at the singular-group level and can achieve better performance on GLUE, specially at higher compression rates.

2. Interesting analysis on the importance score discussing the suitability of using Fisher Information as an importance score.

**Reasons To Reject:**

1. Missing comparison to tensor decomposition techniques [a,b,c], which perform well at high compression rations.

2. No results on generation. There is a rich literature [d,e,f] of matrix decomposition methods for language models and machine translation.




a. https://aclanthology.org/2022.emnlp-main.475/
b. https://aclanthology.org/2022.naacl-main.154.pdf
c. https://proceedings.neurips.cc/paper_files/paper/2019/file/dc960c46c38bd16e953d97cdeefdbc68-Paper.pdf
d. https://proceedings.neurips.cc/paper_files/paper/2018/hash/a2b8a85a29b2d64ad6f47275bf1360c6-Abstract.html
e. https://aclanthology.org/2020.findings-emnlp.250.pdf
f. https://www.isca-speech.org/archive/pdfs/interspeech_2018/shi18_interspeech.pdf

**Reproducibility:**

3: Could reproduce the results with some difficulty. The settings of parameters are underspecified or subjectively determined; the training/evaluation data are not widely available.

**Reviewer Confidence:**

4: Quite sure. I tried to check the important points carefully. It's unlikely, though conceivable, that I missed something that should affect my ratings.

---

> ### Author Rebuttal · Authors · 2023-08-29
>
> Please refer to response to Reviewer **WKgh** for additional results with GPT-2 and BART. We answer all other questions as follows.
>
> > Concern 1. Missing comparison to tensor decomposition techniques [a,b,c], which perform well at high compression rations.
>
> Thanks for sharing these papers. According to the background (section 2) of Reference A:  matrix factorization and tensor-train decomposition are two directions.   Reference A and C seem to align with the concept of tensor-train decomposition, whereas our approach RankDyna aligns more closely with the matrix factorization (SVD, WSVD, and TFWSVD). Reference B is a combination of knowledge distillation (KD) and  matrix-factorization.  Previous work WSVD and TFWSVD  also show that knowledge distillation and matrix-factorization are two orthogonal directions that can be combined.    Our evaluation focus is to examine the effectiveness of SOTA matrix-factorization  methods (SVD, WSVD, and TFWSVD), as they are the most closely related approaches to ours. We are happy to add Reference ABC  to the related work, and thanks again to refer them to us.
>
> > Question 1. In the paper you mention "RankDyna only requires an one-time factorization on the base model, which can be shared across all downstream tasks." Don't you need to apply RankDyna for every downstream task since you would be working on a separate dataset each time?
>
> It is indeed true that we will apply our RankDyna to each downstream task. However, it differs from SVD/WSVD/TFWSVD in that RankDyna achieves compression through a single round of fine-tuning instead of the two rounds typically required by the other methods.
>
> Specifically, the training process for SVD/WSVD/TFWSVD comprises three main phases:
>
> 1. Fine-tuning the general base model to adapt it to the specific downstream task.
> 2. Applying matrix factorization methods such as SVD/WSVD/TFWSVD for compression.
> 3. Fine-tuning the model obtained in step 2 once again.
>
> In contrast, our RankDyna method  only needs one round of fine-tuning, which is essentially equivalent to step 1. This property can benefit storage efficiency. For instance, if we consider ten downstream tasks, the SVD/WSVD/TFWSVD approach will require the storage of ten distinct task-specific models prior to compression. On the contrary, our RankDyna approach only requires the storage of a single general base model. We achieve compact task-specific models by simultaneously conducting fine-tuning and compressing this base model.

---

### Meta-Review · Area_Chair_dHqU · 2023-09-17

**Recommendation:** 3

**Metareview:**

This work proposes a method for compressing Transformer model parameters by leveraging matrix decomposition. Basic idea is to dynamically tracks the importance of parameters and adjust the rank budget to derive a model in fine tuning.

Strengths
* The proposed method is simple yet well-designed by capturing the changes in the importance associated with singular groups.
* Experiments on GLUE show better performance under higher compression rates with detail analysis on the impact of importance score.

Weaknesses
* Experiments are limited to BERT and it has no experiment on other variations. It is not clear whether the proposed approach is also effective for the text generation tasks, e.g., MT.
* Novelty might be limited given the prior work on FWSVD/TFWSVD, i.e., a static variant.

---

### Decision · Program_Chairs · 2023-10-07

**Decision:**

Accept-Findings

**Comment:**

This work proposes a method for compressing Transformer model parameters by leveraging matrix decomposition. Basic idea is to dynamically tracks the importance of parameters and adjust the rank budget to derive a model in fine tuning.

Strengths
* The proposed method is simple yet well-designed by capturing the changes in the importance associated with singular groups.
* Experiments on GLUE show better performance under higher compression rates with detail analysis on the impact of importance score.

Weaknesses
* Experiments are limited to BERT and it has no experiment on other variations. It is not clear whether the proposed approach is also effective for the text generation tasks, e.g., MT.
* Novelty might be limited given the prior work on FWSVD/TFWSVD, i.e., a static variant.